# Investigating student collaborative problem-solving competency and science achievement with multilevel modeling: Findings from PISA 2015

Xuyan Tang[1]*, Yan Liu[2], Marina Milner-Bolotin[3]

1 Department of Educational and Counselling Psychology and Special Education, The University of British Columbia, Vancouver, British Columbia, Canada, 2 Department of Psychology, Carleton University, Ottawa, Ontario, Canada, 3 Department of Curriculum and Pedagogy, The University of British Columbia, Vancouver, British Columbia, Canada

* xuyan.tang@ubc.ca

**Data Availability Statement:** All data are available from the Programme for International Student

## Abstract

Collaborative problem-solving (CPS) competency is critical for 21st century students. However, reports from the Programme for International Student Assessment (PISA) 2015 have revealed significant deficiencies in this competency among young students globally, indicating a critical need for the cultivation of CPS skills. Therefore, it is essential for educators and researchers to examine the factors that influence CPS competency and understand the potential role of CPS in secondary education. The present study aims to investigate the relationship between collaboration dispositions and students' CPS competency as well as the relationships of CPS competency and inquiry-based science instruction (IBSI) with science achievement using the PISA 2015 data. A total of 408,148 students from 52 countries and economies (i.e., regions) were included in our analysis. Unlike most previous studies that only investigated one country at a time and neglected the multilevel data structure of PISA, this study provided a global view through adopting multilevel modeling to account for the cluster effect at the school and country levels. Our findings revealed that valuing relationship was positively associated with CPS, whereas valuing teamwork was negatively associated with CPS. Furthermore, CPS competency was found to be a dominant and positive predictor of science achievement among all study variables, underscoring the importance of integrating CPS into teaching practices to promote student success in science. Additionally, different IBSI activities show varying relationships with science achievement, indicating that caution should be taken when recommending any specific practices associated with IBSI to teachers.

## Introduction

Collaborative problem-solving (CPS) has been increasingly recognized as being critical for succeeding in educational and work settings in the 21st century [1, 2]. The 2015 Programme for International Student Assessment (PISA) defines CPS as "the capacities of an individual to

Assessment (PISA) 2015 database: https://www.oecd.org/pisa/data/2015database/.

**Funding:** The authors received no specific funding for this work.

**Competing interests:** The authors have declared that no competing interests exist.

effectively engage in a process whereby two or more agents attempt to solve a problem by pooling their knowledge, skills and efforts" [3 p.32]. Similarly, the Assessment and Teaching of 21st Century Skills (ATC21S) conceptualizes CPS as "a set of distinguishable subskills which are deployed in accordance with situational needs" [4 p.41]. The Australian Council for Educational Research views it as "a division of labour with participants who are engaged in active discourse that results in a compilation of their efforts" [5 p.2]. Despite the variations in wording to define CPS, it can be seen from these definitions that CPS entails more than simply individuals working together; instead, it involves the synergistic combination of their diverse knowledge, skills, and efforts to reach a solution. As such, CPS competency comprises two important components: the cognitive problem-solving component and the social collaborative component [3, 6].

The evolving complexities associated with modern world problems often challenge students beyond their individual capacities in both content knowledge and problem-solving skills [7]. Consequently, collaboration among individuals with diverse expertise becomes essential, especially in science, technology, engineering and mathematics (STEM) education, as the field itself is highly collaborative and relies on collaborative problem-solve at its core [8]. In a meta-analysis study in STEM contexts, Lai and Wong [9] reviewed 33 publications involving a total of 4717 learners and found that learners who engaged in CPS activities exhibited better cognitive and affective learning outcomes compared to those who engaged in individual problem-solving activities.

However, reports from PISA 2015 have revealed significant deficiencies in CPS competency among young students globally. Only 8% of students from 52 countries achieved the highest level of CPS proficiency, whereas 29% performed at the lowest CPS level [10]. This indicates a critical need for the cultivation of CPS skills in young students and the implementation of relevant educational programs aimed at CPS training. To inform decision making, it is essential to examine the factors that influence CPS competency and understand the potential role of CPS in secondary education before the government and schools make large-scale investments in CPS training across the curriculum. In our literature review, we delve into some major factors related to CPS and how CPS and other important factors are related to secondary school student achievement, specifically in the field of science.

## Collaboration dispositions and CPS competency

Previous studies have found a variety of factors associated with CPS competency, such as social media-related (e.g., attitude towards social media, and purposes and contexts of social media usage), teacher- and school-related (e.g., school location, the proportion of fully certified teachers, and teacher unfairness), and socioenvironmental (e.g., cultures and economic inequality) factors [11–13]. Although some of these factors were shown to be important to CPS development, it is challenging to directly manipulate these external factors in intervention programs of CPS. In contrast, internal factors (e.g., motivation to complete goals, perceived locus of causality, enjoyment of learning [14]) are relatively easy to be included in intervention programs and may be more important to the CPS development because these factors can inherently shape individuals' thinking and behaviors.

To the best of our knowledge, the research examining how internal factors affect CPS performance is currently limited [14]. Based on the existing literature, one such factor worth investigating is collaboration dispositions (i.e., attitudes towards collaboration). Collaboration dispositions have been shown to positively impact a range of outcomes related to attitudes, beliefs, and performance. These outcomes include team cohesiveness, satisfaction with teamwork, expected quality of teamwork, positive thinking, STEM self-efficacy, beliefs about the

values of STEM, as well as perceived learning and academic performance [15–17]. In all these studies, collaboration dispositions were conceptualized as a unidimensional construct, regardless of different measures being used.

In PISA research, however, collaboration dispositions were reflected by two distinct dimensions, including *valuing relationships*, which measures the willingness to interact for altruistic reasons rather than one's own benefit during collaborative activities, and *valuing teamwork*, which represents the extent to which students acknowledge the advantages of collaboration and tend to team up together. The findings from PISA research deviated from previous studies that assumed a unidimensional structure of collaboration dispositions. Two peer-reviewed journal papers (China [11, 18]) and PISA national reports (Canada [19]; England [20]) found that valuing relationships was positively associated with CPS, but valuing teamwork was negatively related to CPS.

However, these findings are solely based on data from three countries (Canada, China, England), and the analytical methods used in PISA research may be problematic. For example, linear regression was employed as the analytical method in these national reports, which fails to take into account the nested structure of PISA data. The nested data structure refers to the scenario in which students from the same school tended to be more similar to each other than students from other schools in the sample (also known as the cluster effect or design effect), so the observations are not independent. Given these limitations, the relationship between collaboration dispositions and CPS needs to be further evaluated with more appropriate statistical models and data from additional countries. Our study aims to fill this gap.

Several demographic variables, including economic, social, and cultural status (ESCS), gender, and Human Development Index (HDI), were included in this study as covariates to control for the confounding effects from these covariates and capture the true relationship between the primary predictors and the outcome variable. ESCS and gender were found to influence CPS performance in PISA 2015 test results, with girls scoring higher than boys in all participating countries [3] and socioeconomically privileged students scoring higher than socioeconomically underprivileged students [19, 20]. In addition, a country-level covariate HDI was retrieved from a source outside the PISA database and added to our analysis to control for the heterogeneity of country characteristics.

## CPS competency, inquiry-based instruction, and science achievement

The integration of CPS training and inquiry-based instruction has been considered as a promising approach to improve student learning outcomes in contemporary science education reform [21, 22]. In scientific fields, CPS and inquiry are closely related, as students apply inquiry skills, such as idea testing, prediction making, and evidence-based reasoning, when engaging in CPS activities [23]. To obtain a further understanding of their roles in science education, we investigated the relationship of CPS competency and inquiry-based science instruction (IBSI) with student science achievement.

## CPS competency and science achievement

It has been argued that "science is best learned when students practice the language and tools of scientific problem-solving in socially situated activities" [24 p.646]. The CPS process would scaffold students to work collaboratively in groups, address conflict situations, apply critical and creative thinking to achieve solutions, and draw conclusions from empirical evidence, which are considered core elements of science literacy [25]. Palincsar et al.'s [24] study has demonstrated an increase in science performance after the implantation of a CPS-based

instructional program, with a sample of 130 middle school students. Additionally, Chan et al.'s [26], Ebrahim's [27], and Musalamani et al.'s [28] studies with sample sizes of 69, 163, and 120 students, respectively, have reported that CPS-based teaching methods have produced significantly more positive effects on student science achievement compared to traditional teaching methods. Although CPS seems a promising factor that may positively influence student science achievement, these findings are either predominantly based on qualitative analysis or derived from quantitative analysis with a small sample size. Recent findings from the PISA 2015 technical report showed that CPS scores had a moderately strong correlation with science scores, ranging from .65 to .83 in different countries [29]. However, to provide further support for the significance of CPS to student science achievement, more studies need to be conducted to examine the relationship between CPS and science achievement.

**IBSI and science achievement.**   Traditional lecture-based learning fails to adequately address students' need of CPS competency in the 21st century. As a result, inquiry-based learning in which "students use an authentic problem as the context for an in-depth investigation of what they need and what to know" is attracting growing attention [30 p.26, 31–33]. Some prior studies employing a quasi-experimental design have consistently shown that inquiry-based teaching yields greater benefits for student science learning than conventional teaching approaches [34–36]. However, studies that used the PISA datasets to examine the relationship between IBSI and science achievement have demonstrated controversial results. When IBSI was used as a composite index variable, students who experienced lower levels of IBSI tended to perform better in science [37–40]. However, both Cairns [37] and Oliver et al. [40] found that individual items of IBSI had varying associations (positive, negative, and almost zero) with science performance. Using factor analysis, two studies found that one factor of IBSI was positively related to science achievement, while the other showed a negative association [41, 42]. Given the complexity of the relationships between IBSI and science achievement, individual item scores of IBSI were used in our study to examine how specific activities or practices of IBSI influenced science achievement scores.

In addition to CPS and IBSI, four attitudinal and three demographic factors were included in our study as covariates to adjust for their effects on the outcome variable and increase the precision of estimates. Attitudinal factors comprised a set of variables that reflected attitudes toward science measured by the PISA's student questionnaire, including science self-efficacy, enjoyment of science, interest in broad science topics, and epistemological beliefs. These factors were all found to be positively associated with PISA science achievement [43–45]. For demographic variables, ESCS had a positive correlation with PISA science achievement across countries [40] and boys had higher scores than girls overall in the PISA 2015 science assessment [46]; hence, ESCS and gender must be controlled for. HDI was also used to adjust for the heterogeneity of country characteristics.

In order to uncover the key features of developing CPS and science competencies around the globe, we performed a comprehensive analysis using the PISA 2015 dataset. We used multilevel models to examine the effect of student- and country-level variables, within and between variations across countries, and addressed two research questions:

RQ-1: How are dispositions towards collaboration (valuing relationships and valuing teamwork) associated with students' CPS competency?

RQ-2: How are CPS competency and IBSI related to students' science achievement?

Of note, for RQ-1, gender, ESCS, and HDI were included to control for gender difference and differences in the individual's and country's socioeconomic status. For RQ-2, four more covariates (i.e., science self-efficacy, enjoyment of science, interest in broad science topics, and

epistemological beliefs) were added to control for individual differences in their beliefs in and attitudes towards science.

## Methods

### Data source and sample

The data used in this study were retrieved from the PISA 2015 (https://www.oecd.org/pisa/data/2015database/), which is a large-scale, international assessment that evaluated both CPS competency and science achievement. It should be noted that the CPS assessment was conducted only in the PISA 2015 cycle. A two-stage stratified sampling design was adopted in PISA data collection to ensure the representativeness of the sample, with schools being sampled using probability proportional to their size (i.e., the estimated number of 15-year-olds enrolled in PISA-eligible schools), and students being sampled with equal probability within schools [29]. Out of 72 countries and economies (i.e., regions) that participated in PISA 2015, 20 did not take part in the CPS assessment due to its optional nature and requirement of computer access. Hence, this study only included 52 countries and economies with a total sample of 408,148 students. Ethics approval was not required for this study, as the extracted data were anonymized before access and analysis.

### Measures

**Predictors.** All measures used from the PISA 2015 student questionnaire were listed in S1 File. For collaboration dispositions, the *valuing relationships* index was derived from four items rated on a 4-point Likert-type scale and the *valuing teamwork* index was derived from another set of four items. The range of Cronbach's alpha reliability estimates was 0.59–0.75 for valuing relationships and 0.57–0.87 for valuing teamwork across countries [29]. Students' total scores of each scale were transformed using a generalized partial credit model and standardized to a metric with a mean of 0 and a standard deviation of 1, with higher scores indicating more positive collaboration dispositions. For more detailed descriptions of data transformation, please refer to the PISA technical report [29]. A total of nine items were included in the IBSI scale and each was used as an individual predictor. All items were rated on a 4-point Likert scale and were reverse scored, with higher values representing a higher frequency of activities in science lessons. This scale demonstrated good reliability across countries, with Cronbach's alpha ranging from 0.83 to 0.90 [29].

**Covariates.** Seven covariates were included, i.e., gender, HDI, ESCS, science self-efficacy, enjoyment of science, interest in broad science topics, and epistemological beliefs. Gender was included as a control variable (0 = girls; 1 = boys). *HDI* was a composite index of life expectancy, education, and per capita income indicators, which reflected the social and economic development levels of countries [47]. It was ranked on a scale from 0 to 1. *ESCS* was a composite score derived from three indices concerning the economic, social, and cultural status: home possessions, highest parental occupational status, and highest parental educational level. The Cronbach's alpha for this measure ranged from 0.36 to 0.77 across countries [29]. For the attitude variables, the scales of *science self-efficacy*, *enjoyment of science*, *epistemological beliefs*, and *interest in broad science topics* consisted of eight, five, six, and five items, respectively. The four measures showed acceptable reliability across countries, with Cronbach's alpha of 0.83–0.94, 0.85–0.97, 0.80–0.94, and 0.72–0.86, respectively [29]. All these scales were rated on a 4-point scale except a 5-point scale of interest in broad science topics. The index of ESCS and attitudes toward science was standardized with a mean of 0 and a standard deviation of 1 by PISA researchers. Additional details about the transformation process can be found in the PISA technical report [29].

**Outcomes.** CPS and science scores were the outcome variable in RQ-1 and RQ-2, respectively. CPS scores were also used as a predictor to address RQ-2. In the PISA 2015 CPS assessment, students collaborated with computer-simulated agents to complete various tasks involving real-life problem scenarios [10]. They made selections from lists of predefined messages and earned credits when choosing the correct answer. Aside from message communication, non-chat problem-solving actions (e.g., moving elements on the screen) were also recorded and scored as a reflection of CPS competency. With the use of predefined stimuli, PISA 2015 outlined the expected interaction patterns and collaborative behaviors, thereby providing standardized assessment conditions and ensuring the comparability of CPS scores.

Several validation studies have shown that PISA 2015 CPS is "functioning reasonably well" [48 p.12]. Scoular et al. [48] demonstrated that the CPS test was measuring an assumed unidimensional collaboration construct using item response theory analysis on PISA data. Additionally, they found the test to have a good coverage of items, with most showing moderate difficulty level, and be sufficiently sensitive to differentiate students with varying collaborative abilities. Other empirical studies applying released PISA 2015 CPS tasks also provided validity evidence. Herborn et al. [49] found no significant differences in CPS performance accuracy between students collaborating with human partners and those with computer agents, supporting the use of computer-simulated agents in collaborative environments. Stadler et al. [50] revealed that student CPS scores were moderately related to their self-rated scores on collaborative abilities and teacher-rated scores on student collaborative and reasoning abilities. Moreover, CPS scores strongly predicted students' collaboration performance with their own team player [50]. Together, these studies supported that this test was assessing the CPS construct as intended by the PISA developers.

In the 2015 cycle, there were 184 items in the science assessment and 117 items in the CPS assessment, and each student only answered part of them due to time constraints. For CPS and science assessments, 10 plausible values (PVs) ranging from 1 to 1000 were estimated for each student to represent their overall competencies by PISA researchers, using item response theory scaling methods [29]. The traditional approach for handling PVs was to use only one of them or an average of all PVs, resulting in a limited amount of information being incorporated. To fully utilize all the information provided by PVs, it is recommended to adopt Aparicio et al.'s [51] approach, which was to analyze each model 10 times with each of the PVs and report an average of 10 sets of estimates. This approach allows for a more accurate representation of educational outcomes, as it uses the whole distribution of test scores as a proxy of student performance [51]. Following this recommendation, we used 10 CPS PVs for the outcome in RQ-1. However, in RQ-2 we encountered the challenge of having multiple PVs for both the predictor variable (i.e., CPS) and the outcome variable (i.e., science achievement), which adds complexity to data analysis. As a result, we incorporated 10 science PVs for the outcome but used the mean of 10 CPS PVs for the predictor variable to simplify the analysis.

Given that the analysis included variables with varying ranges (e.g., 0–1, 1–1000), a scaling technique was applied to ensure that all variables were on a comparable scale, facilitating meaningful interpretations and helping with model convergence. Specifically, the CPS mean was scaled down by a factor of 100 (e.g., a CPS mean of 668.04 in raw data was transformed to 6.68 for analysis) and the HDI was scaled up by a factor of 10 (e.g., an HDI index of 0.92 in raw data was transformed to 9.2 for analysis) to mitigate potential discrepancies in magnitudes.

## Data analysis

Multilevel modeling was performed using R version 4.0.4 [52] with the *merTools* R package [53]. To account for the cluster effect resulted from the sampling design in PISA data

collection, three-level models with random intercepts were employed. To take account of 10 PVs for the outcome in both research questions, we used *lmerModList()* function in *merTools* R package to provide the final estimates from 10 sets of data analyses. We followed the model building strategies suggested by Raudenbush and Bryke [54] and added variables sequentially. For RQ-1, we started from the unconditional means model (Model-1A), entered the country-level covariate (Model-1B), subsequently added the student-level covariates (Model-1C), and eventually built the full model (Model-1D) by adding the predictors of interest. The equation of the full model is presented as follows:

$$
\begin{aligned}
CPS_{ijk} = \quad & \gamma_{000} + \gamma_{001}\ HDI + \gamma_{100}\ gender + \gamma_{200}\ ESCS + \gamma_{300}\ valuing\ relationships \\
& + \gamma_{400}\ valuing\ teamwork + e_{ijk} + r_{0jk} + u_{00k}
\end{aligned}
\tag{1}
$$

where $CPS_{ijk}$ denotes the outcome variable; $\gamma_{000}$ denotes the intercept, i.e., the average CPS competency across all schools and all countries in the sample; $\gamma_{001} - \gamma_{400}$ are the regression coefficients for HDI, gender, ESCS, valuing relationships, and valuing teamwork; three random effects are included in the analysis: $e_{ijk}$ representing the deviation of student $I$ score from the school mean score, $r_{0jk}$ representing the deviation of school $j$ mean score from the country mean, and $u_{00k}$ representing the deviation of country $k$ mean score from the grand mean.

Similarly, for RQ-2, we started from the unconditional means model (Model-2A), added both the country- and student-level covariates (Model-2B), followed by the addition of IBSI (Model-2C), and eventually entered CPS competency (Model-2D). The equation of the final model is presented as follows:

$$
\begin{aligned}
Science_{ijk} = \quad & \gamma_{000} + \gamma_{001}\ HDI + \gamma_{100}\ gender + \gamma_{200}\ ESCS + \gamma_{300}\ beliefs + \gamma_{400}\ enjoyment + \\
& + \gamma_{500}\ interest + \gamma_{600}\ self\_efficacy + \gamma_{700}IBSI\,1 + \gamma_{800}IBSI\,2 \\
& + \gamma_{900}IBSI\,3 + \gamma_{1000}IBSI\,4 + \gamma_{1100}IBSI\,5 + \gamma_{1200}IBSI\,6 + \gamma_{1300}IBSI\,7 \\
& + \gamma_{1400}IBSI\,8 + \gamma_{1500}IBSI\,9 + \gamma_{1600}CPS + e_{ijk} + r_{0jk} + u_{00k}
\end{aligned}
\tag{2}
$$

where $Science_{ijk}$ is the outcome variable; $\gamma_{001} - \gamma_{1600}$ are the regression coefficients of HDI, gender, ESCS, epistemological beliefs, enjoyment of science, interest in broad science topics, science self-efficacy, and IBSI on science achievement. IBSI 1 to IBSI 9 represent the individual items from the IBSI questionnaire.

Missing values in the dataset were imputed using random forests algorithm, as implemented in the R package *missRanger* [55]. This imputation approach has shown low imputation errors and similar performance as multiple imputation [56].

## Results

Descriptive statistics of all the variables used in this study are presented in Table 1.

Before running the full scale of data analyses, we conducted a small pilot study to examine whether the relationships we tested in the present study were held across countries. We selected seven countries with low, medium, and high CPS performance on the PISA tests and conducted the same models described above for each country. The results showed consistent patterns across these seven countries, which were similar to the findings based on 52 participating countries and economies. This pilot study supported our use of all-country data in the present study. Additionally, the multilevel modeling allows the outcome to vary across schools and countries. The results with 52 countries and economies are presented below.

**Table 1. Descriptive statistics of variables used in the present study.**

| Continuous Variable | Mean | SD | Range |
|---|---|---|---|
| CPS competency | 485.93 | 92.19 | 123.79–893.23 |
| Science achievement | 479.99 | 97.23 | 133.43–835.62 |
| Valuing relationship | 0.05 | 0.99 | -3.33–2.29 |
| Valuing teamwork | 0.10 | 0.98 | -2.83–2.14 |
| IBSI 1 | 2.92 | 0.94 | 1–4 |
| IBSI 2 | 1.93 | 0.86 | 1–4 |
| IBSI 3 | 2.11 | 0.94 | 1–4 |
| IBSI 4 | 2.33 | 0.94 | 1–4 |
| IBSI 5 | 2.7 | 0.93 | 1–4 |
| IBSI 6 | 1.74 | 0.93 | 1–4 |
| IBSI 7 | 1.98 | 0.96 | 1–4 |
| IBSI 8 | 2.58 | 0.98 | 1–4 |
| IBSI 9 | 2.08 | 0.96 | 1–4 |
| Epistemological beliefs | -0.02 | 1.00 | -2.79–2.16 |
| Enjoyment of science | 0.11 | 1.10 | -2.12–2.16 |
| Interest in broad science topics | 0.09 | 0.98 | -2.58–2.73 |
| Science self-efficacy | 0.05 | 1.25 | -3.76–3.28 |
| ESCS | -0.22 | 1.09 | -7.26–4.07 |
| HDI | 0.85 | 0.07 | 0.73–0.95 |
| Continuous Variable | Frequency (%) | | |
| Gender | Girls: 204,642 (50.1) | | Boys: 203,506 (49.9) |

*Note*. HDI = Human Development Index; ESCS = Economic, Social, and Cultural Status; IBSI = Inquiry-based Science Instruction; CPS = Collaborative Problem-Solving. CPS competency in this table is the mean of 10 CPS plausible values, and science achievement in this table is the mean of 10 science plausible values.

## RQ 1: How are collaboration dispositions (valuing relationships and valuing teamwork) related to students' CPS competency?

Model-1A to D were used to address RQ-1, and the results are provided in Table 2. The unconditional means model (Model-1A) showed that 21% of the total variance in students' CPS competency was accounted for by school differences with an intra-class correlation (ICC) of 0.21, and 17% accounted by country differences (ICC = 0.17). The results supported the application of multilevel models to take account of the cluster effect in the data.

Model-1B examined the role of the country-level covariate. With 0.1 unit increase in the HDI index, student CPS scores were predicted to increase on average by 43.56 points. Results on variance reduction indicated that HDI explained 49% of the country-level variance in students' CPS competency. The effect of student-level covariates was explored in Model-1C. Student ESCS was positively related to their CPS scores, and female students scored 19.50 points higher than male students after controlling for the effect of HDI and other variables. These student-level covariates (i.e., gender and student-level ESCS) explained 3% of the student-level variance and 25% of the school-level variance. The literature has shown that the school-level variance can be explained by adding a student-level variable, if its mean varies over schools [57].

The key predictors, valuing relationships and valuing teamwork, were added to the full model (Model-1D). They showed opposite effects on CPS competency. After controlling for other variables, valuing relationship was positively related to CPS scores ($\gamma_{300} = 14.75$),

**Table 2. The analysis results of Model-1A to Model-1D.**

| Fixed effects | Unconditional Means Model (Model-1A) | | | Model-1B | | | Model-1C | | | Full model (Model-1D) | | |
|---|---|---|---|---|---|---|---|---|---|---|---|---|
| | Parameter Estimates | SE | t | Parameter Estimates | SE | t | Parameter Estimates | SE | t | Parameter Estimates | SE | t |
| Intercept | 484.56* | 5.89 | 82.30 | 483.79* | 4.22 | 114.65 | 499.26* | 4.24 | 117.79 | 497.69* | 4.26 | 116.71 |
| Country level | | | | | | | | | | | | |
| HDI | | | | 43.56* | 6.26 | 6.96 | 34.88* | 6.29 | 5.55 | 34.35* | 6.33 | 5.43 |
| Student level | | | | | | | | | | | | |
| ESCS | | | | | | | 15.39* | 0.17 | 91.59 | 14.04* | 0.17 | 84.34 |
| Gender | | | | | | | -22.96* | 0.32 | -71.00 | -19.50* | 0.33 | -59.40 |
| Valuing Relationship | | | | | | | | | | 14.75* | 0.15 | 99.20 |
| Valuing Teamwork | | | | | | | | | | -10.63* | 0.16 | -64.83 |
| *Random effects* | Variance Component | | | Variance Component | | | Variance Component | | | Variance Component | | |
| Student level | 6441.03 | | | 6441.03 | | | 6233.73 | | | 6068.10 | | |
| School level | 2237.20 | | | 2237.10 | | | 1676.49 | | | 1599.20 | | |
| Country level | 1739.06 | | | 879.54 | | | 887.27 | | | 899.04 | | |
| AIC | 4770300 | | | 4770270 | | | 4753820 | | | 4742590 | | |
| *Variance reduction* | | | | | | | | | | | | |
| Student level | | | | 0 | | | 0.03 | | | 0.03 | | |
| School level | | | | 0 | | | 0.25 | | | 0.05 | | |
| Country level | | | | 0.49 | | | -0.01 | | | -0.01 | | |

*Note.* HDI = Human Development Index; ESCS = Economic, Social, and Cultural Status.

* $p < .001$

whereas valuing teamwork was negatively associated with CPS scores ($\gamma_{400} = 19.50$). Together, collaboration dispositions explained 3% of the student-level variance and 5% of the school-level variance in CPS competency on the top of the variance explained by other covariates. The results of the full model showed that students with higher ESCS and higher HDI had achieved higher CPS scores and female students had better CPS competency than males.

### RQ2. How are CPS competency and IBSI related to students' science achievement?

Table 3 presents the results of Model-2A to D. The finding of the unconditional mean model (Model-2A) demonstrated the necessity of multilevel modeling, because 27% of the variance in students' science achievement was accounted for by school differences (ICC = 0.27) and 16% accounted for by country differences (ICC = 0.16). Model-2B shows that all the covariates are significantly related to science achievement. Students from countries with higher HDI ($\gamma_{001} = 31.31$) performed better on the PISA science assessment. In terms of student-level covariates, male students ($\gamma_{100} = 4.79$) and students with higher ESCS ($\gamma_{200} = 13.16$) and more positive attitudes toward science, i.e., epistemological beliefs ($\gamma_{300} = 16.65$), enjoyment of science ($\gamma_{400} = 8.60$), interest in broad science topics ($\gamma_{500} = 6.73$), and science self-efficacy ($\gamma_{600} = 4.70$) had higher science scores.

Nine IBSI variables were added to Model-2C, which showed varying relationships with science achievement, and all of them were statistically significant. Four of these IBSI activities showed to be positive predictors of science achievement, including *students explaining*

**Table 3. The analysis results of Model-2A to Model-2D.**

| | Unconditional Means Model (Model-2A) | | | Model-2B | | | Model-2C | | | Full Model (Model-2D) | | |
|---|---|---|---|---|---|---|---|---|---|---|---|---|
| *Fixed effects* | Parameter Estimates | SE | t | Parameter Estimates | SE | t | Parameter Estimates | SE | t | Parameter Estimates | SE | t |
| Intercept | 477.02* | 5.77 | 82.61 | 477.12* | 4.42 | 107.91 | 501.40** | 4.06 | 123.45 | 473.80** | 1.65 | 286.83 |
| Country level | | | | | | | | | | | | |
| HDI | | | | 31.31* | 6.56 | 4.77 | 27.19** | 5.93 | 4.58 | 0.17 | 2.36 | 0.07 |
| Student level | | | | | | | | | | | | |
| Gender | | | | 4.79* | 0.25 | 19.13 | 8.33** | 0.24 | 34.38 | 25.06** | 0.17 | 150.47 |
| ESCS | | | | 13.16* | 0.15 | 90.76 | 13.20** | 0.14 | 94.26 | 4.08** | 0.10 | 41.91 |
| Epistemological beliefs | | | | 16.65* | 0.13 | 127.14 | 14.77** | 0.13 | 116.00 | 4.34** | 0.09 | 48.43 |
| Enjoyment of science | | | | 8.60* | 0.14 | 61.53 | 8.76** | 0.14 | 64.34 | 4.78** | 0.09 | 50.84 |
| Interest in broad science topics | | | | 6.73* | 0.15 | 45.11 | 6.80** | 0.14 | 47.11 | 2.75** | 0.10 | 28.01 |
| Science self-efficacy | | | | 4.70* | 0.10 | 46.43 | 6.22** | 0.10 | 63.22 | 3.89** | 0.07 | 57.97 |
| IBSI 1 | | | | | | | 0.54** | 0.14 | 3.82 | -0.24* | 0.10 | -2.45 |
| IBSI 2 | | | | | | | -3.00** | 0.18 | -16.66 | -0.38* | 0.12 | -3.06 |
| IBSI 3 | | | | | | | -4.52** | 0.16 | -27.69 | -0.68** | 0.11 | -6.19 |
| IBSI 4 | | | | | | | 2.85** | 0.17 | 16.92 | 0.62** | 012 | 5.37 |
| IBSI 5 | | | | | | | 9.55** | 0.16 | 59.73 | 2.07** | 0.11 | 18.86 |
| IBSI 6 | | | | | | | -11.56** | 0.17 | -69.35 | -2.39** | 0.11 | -20.86 |
| IBSI 7 | | | | | | | -7.30** | 0.17 | -42.56 | -1.70** | 0.12 | -14.32 |
| IBSI 8 | | | | | | | 2.77** | 0.15 | 18.34 | 0.47** | 0.10 | 4.55 |
| IBSI 9 | | | | | | | -8.13** | 0.17 | -47.59 | -2.33** | 0.12 | -19.81 |
| CPS | | | | | | | | | | 81.97** | 0.12 | 681.59 |
| *Random effects* | Variance Component | | | Variance Component | | | Variance Component | | | Variance Component | | |
| Student level | 5778.74 | | | 4947.01 | | | 4607.83 | | | 2076.26 | | |
| School level | 2797.25 | | | 1795.13 | | | 1510.41 | | | 330.48 | | |
| Country level | 1675.10 | | | 979.44 | | | 799.53 | | | 124.10 | | |
| AIC | 4730180 | | | 4663130 | | | 4430620 | | | 4299080 | | |
| *Variance reduction* | | | | | | | | | | | | |
| Student level | | | | 0.14 | | | 0.07 | | | 0.55 | | |
| School level | | | | 0.36 | | | 0.16 | | | 0.78 | | |
| Country level | | | | 0.42 | | | 0.18 | | | 0.84 | | |

*Note.* HDI = Human Development Index; ESCS = Economic, Social, and Cultural Status; IBSI = Inquiry-based Science Instruction; CPS = Collaborative Problem-Solving.

** $p < .001$,

* $p < .05$

their ideas ($\gamma_{700} = 0.54$), *students drawing conclusions from experiments* ($\gamma_{1000} = 2.85$), *teachers explaining applications of science ideas* ($\gamma_{1100} = 9.55$), *and teachers explaining relevance of science concepts* ($\gamma_{1400} = 2.77$). However, the other five activities showed a negative association with science achievement, including *spending time in the laboratory doing experiments* ($\gamma_{800} = -3.00$), *arguing about science questions* ($\gamma_{900} = -4.52$), *designing their own experiments* ($\gamma_{1200} = -11.56$), *having class debates about investigations* ($\gamma_{1300} = -7.30$), and *doing investigations to test ideas* ($\gamma_{1500} = -8.13$). The student-, school-, and country-level variance in science achievement was further accounted for by IBSI variables with a reduction of 7%, 16% and 18%, respectively.

Finally, CPS was added to the full model (Model-2D), which was used to examine the effect of CPS competency on science achievement after controlling for all other variables. Among all variables, CPS was shown to be a dominant predictor; every 100-point increase in CPS was accompanied by an increase of 81.97 points in science achievement. It explained more than half of the student- (55%), school- (78%), and country-level (84%) variance in science achievement on the top of the variances explained by all other variables. While all the student-level covariates still had significant relationships with the science achievement, HDI was no longer significant, and the effect of IBSI 1 (*students explaining their ideas*) changed from positive ($\gamma_{700}$ = 0.54) to negative ($\gamma_{700}$ = -0.24) in the presence of CPS.

## Discussion and conclusions

The present study investigated the associations of collaboration dispositions (valuing teamwork and valuing relationship) with CPS as well as the relationships of CPS and IBSI with science achievement using the PISA 2015 dataset. Our results showed that valuing relationship was positively related to CPS, whereas valuing teamwork was negatively related to CPS. Additionally, higher CPS competency corresponded to higher science achievement, and different IBSI activities showed varying relationships with science achievement. Our findings contributed to the existing empirical evidence in two critical areas. First, this study offers a global view on important factors influencing CPS competency and science achievement by utilizing large-scale data from 52 countries and economies. Second, this study provides a comprehensive overview of the impact of these factors through the use of multilevel modeling, taking into account the cluster effect at the school and country levels. A thorough discussion of our findings is presented below.

### The relationship between collaboration dispositions and students' CPS competency

Results of the present study suggest that valuing relationships with others were positively associated with students' CPS competency, such as *considering different perspectives*, *being a good listener*, *taking into account what others are interested in*, and *enjoying seeing classmates be successful*, whereas valuing teamwork for its benefits were negative associated with CPS competency, such as *raising one's own efficiency* and *making better decisions*. These findings are in line with previous studies that utilized PISA 2015 data [11, 18–20]. It is noteworthy that, while valuing teamwork and valuing relationships constituted integral components of students' collaboration dispositions in PISA 2015, these two dimensions had a moderate correlation ($r$ = 0.45) in our sample, indicating distinctions in their conceptualization. Expanding on the concept of valuing teamwork, although it provides certain advantages and can increase individual willingness to collaborate, it does not necessarily lead to more significant contributions to successful problem-solving. Effective collaboration requires shared responsibility and a dynamic distribution of labour [48]. Team members who exhibit reduced effort in group work (i.e., social loafing) [58] or lack competency [23] can dramatically impair the overall team performance, as it places additional burdens on other team members to compensate and strive towards achieving team goals. Without active and equal participation of each member, emphasizing the benefits of teamwork may have the opposite effect [59]. Further empirical studies are warranted to better understand the complex relationships between these collaboration dispositions and CPS as well as the underlying processes at play.

Furthermore, the beneficial impact of valuing relationships reveals the significance of social interaction quality within the CPS context, as indicated by extant literature [60, 61]. In addition to being active participants, students need to acquire a range of social skills, such as

perspective-taking, active listening, negotiation, and conflict resolution, to communicate effectively in a team and make sense of problems collectively [62]. Such findings have some practical implications for educators aiming to designing effective intervention programs. First, teachers are recommended to enhance students' motivation and engagement by showing them the relevance and importance of collaboration in task success. Second, teachers should incorporate classroom activities focusing on facilitating students' communication, negotiation, as well as group and task management skills. By offering students abundant opportunities to practice and providing constructive feedback their peer interactions, teachers can improve current levels of individual collaboration, which leads to greater learning outcomes.

## The relationships of CPS competency and IBSI with students' science achievement

Another important finding is that CPS competency was a dominant and positive predictor of science achievement in PISA 2015, suggesting that CPS is conducive to science learning [63]. The cognitive aspect of CPS encompasses essential skills for scientific inquiry such as planning, executing, and monitoring, flexibility, and knowledge building, while the social aspect involves collaboration skills such as participation, perspective taking, and social regulation [4]. With the cognitive skills to analyze and solve problems and social skills to collaborate and communicate, students can leverage their collective knowledge and skills to tackle scientific questions [14]. As such, students with higher CPS competency can have better performance in science. It is worth noting that, unlike science, CPS is rarely taught as a school subject in most countries [64], although there have been some instructional programs specifically designed for fostering CPS [23, 60, 65]. In order to benefit science learning outcomes, it is recommended to integrate CPS as a learning practice in classroom and incorporate it into student work. Beyond the realm of science education, the advantage of implementing CPS can extend to other disciplines where problem-solving plays a central role. For instance, existing literature has demonstrated the positive outcomes of implementing CPS-based programs in STEM education [66, 67].

The relationship between IBSI and science achievement was found to vary depending on the particular activity being examined in this study. Four out of the nine activities had a positive effect on students' performance in science, while the remaining activities had a negative effect. These two groups of items found are largely consistent with the two main forms of instructional practices that prior research identified within the IBSI framework: guided inquiry, which involves some kind of teacher guidance, and independent inquiry, which is student-led [41, 42]. Although there were slight differences in the specific items that fell under guided inquiry and independent inquiry between the current study and Aditomo and Klieme's [41] study, our results support their conclusion that guided inquiry was positively related to science learning outcomes while independent inquiry was negatively related.

It is worth noting, however, that one should not interpret our results as implying that independent inquiry impedes science learning. One possible explanation for the negative relationships found in this study is that students from some countries have rarely, if ever, experienced such activities as argumentative discussions and debates in science classrooms [68, 69]. Hence, their ratings of occurrence of these activities were low, even though they had relatively high scores on the science assessment. Another possible reason is that the PISA 2015 science assessment has been designed to evaluate student performance on a much broader scope (e.g., chemistry, biology, physics, earth and space science), which requires knowledge that goes beyond lab experiments and scientific investigations. For example, one of the released PISA 2015 science items asks about the reason why a meteoroid speeds up when it approaches Earth and its

atmosphere [70]. In the case of this astronomy problem, experiment and investigation activities may not be helpful experiences for students in answering the question.

Nonetheless, there are certain IBSI activities that have showed consistent positive effects in this study and prior research [37, 41], such as teachers explaining the applications of science ideas, teachers explaining relevance of science concepts, and students explaining their ideas. It is recommended that future research should examine the effects of each IBSI practice on science achievements with experimental designs and provide more robust evidence for the use or avoidance of particular practices in science education.

The present study is not without limitations. First, we included predictors of interest based on existing literature, but we might have missed other significant variables that are not yet identified. In particular, a substantial portion of the variation in students' CPS and science scores was attributed to school-level and country-level differences; however, our analysis only included one variable at the country level (i.e., HDI) and did not incorporate any at the school level. It is necessary to add school- and country-level factors to explore their effects on CPS and science achievement in future research. Second, the PISA 2015 CPS assessment is fully computer-based, and therefore, the lack of access to modern technology might limit student participation in certain countries, especially those with the low HDI, and thus might limit the generalizability of our findings. The extent to which our findings can be generalized to countries that did not participate in PISA 2015 CPS need to be examined in the future. Lastly, our study does not provide evidence for cause-effect relationships. Experimental studies are encouraged to investigate what factors can improve CPS skills and how CPS can improve student learning outcomes. Our study can serve as a starting point for researchers to delve into the development of CPS and science competencies and inform educators who want to design instructional practices that aims at facilitating these essential skills among secondary school students.

## Supporting information

**S1 File. Measures selected from PISA2015 student questionnaire for the present study.** (DOCX)

## Acknowledgments

We acknowledge the Programme for International Student Assessment (PISA) 2015 for providing access to the valuable data used in this study. The availability of such a comprehensive and well-curated dataset has been instrumental in conducting our research. We are grateful for the opportunity to utilize this dataset and acknowledge its significant contribution to the outcomes of our study.

## Author Contributions

**Conceptualization:** Xuyan Tang, Yan Liu.

**Formal analysis:** Xuyan Tang.

**Methodology:** Xuyan Tang, Yan Liu.

**Project administration:** Xuyan Tang.

**Software:** Xuyan Tang.

**Supervision:** Yan Liu.

**Writing – original draft:** Xuyan Tang.

**Writing – review & editing:** Xuyan Tang, Yan Liu, Marina Milner-Bolotin.

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
