## [Decision Letter · Decision Letter 0]

15 Sep 2023

PONE-D-23-22154Investigating student collaborative problem-solving competency and science achievement with multilevel modeling: Findings from PISA 2015PLOS ONE

Dear Dr. Tang,

Thank you for submitting your manuscript to PLOS ONE. After careful consideration, we feel that it has merit but does not fully meet PLOS ONE’s publication criteria as it currently stands. Therefore, we invite you to submit a revised version of the manuscript that addresses the points raised during the review process.

Please submit your revised manuscript by Oct 30 2023 11:59PM. If you will need more time than this to complete your revisions, please reply to this message or contact the journal office at plosone@plos.org. Please include the following items when submitting your revised manuscript:A rebuttal letter that responds to each point raised by the academic editor and reviewer(s). You should upload this letter as a separate file labeled 'Response to Reviewers'.A marked-up copy of your manuscript that highlights changes made to the original version. You should upload this as a separate file labeled 'Revised Manuscript with Track Changes'.An unmarked version of your revised paper without tracked changes. You should upload this as a separate file labeled 'Manuscript'.If applicable, we recommend that you deposit your laboratory protocols in protocols.io to enhance the reproducibility of your results. Protocols.io assigns your protocol its own identifier (DOI) so that it can be cited independently in the future. For instructions see: https://journals.plos.org/plosone/s/submission-guidelines#loc-laboratory-protocols. Additionally, PLOS ONE offers an option for publishing peer-reviewed Lab Protocol articles, which describe protocols hosted on protocols.io. Read more information on sharing protocols at https://plos.org/protocols?utm_medium=editorial-email&utm_source=authorletters&utm_campaign=protocols.

We look forward to receiving your revised manuscript.

Kind regards,

Sonia Vasconcelos, PhD

Academic Editor

PLOS ONE

Journal Requirements:

**Additional Editor Comments:**

In your review, pay particular attention to the comments and points raised by Reviewer 1.

Reviewers' comments:

Reviewer's Responses to Questions

**Comments to the Author**

1. Is the manuscript technically sound, and do the data support the conclusions?

Reviewer #1: Yes

Reviewer #2: Yes

2. Has the statistical analysis been performed appropriately and rigorously? 

Reviewer #1: I Don't Know

Reviewer #2: Yes

3. Have the authors made all data underlying the findings in their manuscript fully available?

Reviewer #1: Yes

Reviewer #2: Yes

4. Is the manuscript presented in an intelligible fashion and written in standard English?

Reviewer #1: Yes

Reviewer #2: Yes

5. Review Comments to the Author

Reviewer #1: The present study evaluates the contribution of collaborative problem-solving (CPS) competency to secondary education. By analyzing the Programme for International Student Assessment (PISA 2015), the authors examined factors that influence CPS competency and the relation of inquiry-based science instruction (IBSI) with science achievement. Data from 52 countries were included in the study providing a global view on the factors that matters for CPS and science achievements. Interestingly the data showed that CPS presented positive correlation with science achievements pointing out to the importance of integration CPS into teaching practices. Regarding IBSI, different activities performed by the teachers have positive and negative correlation with science performance among the students.

The studies present limitations, and some of them are clearly presented in the discussion section by the authors. But, as stated, as a starting point to address this important topic, the present study is a very good beginning.

Minor points:

-Author should include schemes in the discussion section to summarize their results, such as what practices/activities correlates positive/negative with CPS and IBSI.

-Is it possible to have an overview on each country data and how they correlate with CPS/IBSI and science achievements? Is there any outlier in the analysis?

-It is sometimes hard to follow how raw data were converted into the data presented in the models. Mainly to the ones who do not know about statistics but have interest in this subject and in science education.

-Do the authors think that CPS could have impact in other disciplines?

-It is not very clear why there are differences between valuing relationships and valuing teamwork to outcomes of the present investigation. They seem to correlate to each other.

Reviewer #2: The authors have done a fully revision of the manuscript. The authors have attended to the concerns highlighted by the reviewers. Although the authors have not provided the contents of the REU program, they addressed the Vygotsky's social constructivism theory.

The study was done with PISA 2015 data. Using the multilevel data the study aimed to investigate the relationship between collaboration dispositions and students’ collaborative problem-solving (CPS) competency. Also, CPS competency and inquiry-based science instruction were related to science achievement. Some novel data were highlighted by controlling some of the multilevel factors. For example, the study unveiled an important data that was masked by the HDI (countries indexes) effect, so that for some countries female students scores are higher than male students scores, showing that female students have more CPS competency than males.

6. PLOS authors have the option to publish the peer review history of their article (what does this mean?). If published, this will include your full peer review and any attached files.

Reviewer #1: No

Reviewer #2: **Yes: **Andréa Carla de Souza Góes

---

## [Author Response · Author response to Decision Letter 0]

14 Oct 2023

Dear Dr. Sonia Vasconcelos,

Thank you for your letter. The comments and suggestions provided by the editor and reviewers have significantly strengthened our paper. We greatly appreciate the editors’ and reviewers’ efforts. In response to the feedback we received, we have carefully addressed each comment below and made the necessary changes to the original manuscript. Any revisions resulting from these comments are clearly shown in blue font in the manuscript. Additionally, we changed APA format to Vancouver format and revised all the citations and the “References” section. Given that this is just a format change, we did not use blue highlight. 

Reviewer #1: 

The present study evaluates the contribution of collaborative problem-solving (CPS) competency to secondary education. By analyzing the Programme for International Student Assessment (PISA 2015), the authors examined factors that influence CPS competency and the relation of inquiry-based science instruction (IBSI) with science achievement. Data from 52 countries were included in the study providing a global view on the factors that matters for CPS and science achievements. Interestingly the data showed that CPS presented positive correlation with science achievements pointing out to the importance of integration CPS into teaching practices. Regarding IBSI, different activities performed by the teachers have positive and negative correlation with science performance among the students.

The studies present limitations, and some of them are clearly presented in the discussion section by the authors. But, as stated, as a starting point to address this important topic, the present study is a very good beginning.

Responses: We appreciate your thoughtful evaluation of our work and your positive feedback. We have made some changes to improve the quality of our manuscript as per your suggestions.

Minor points:

-Author should include schemes in the discussion section to summarize their results, such as what practices/activities correlates positive/negative with CPS and IBSI.

Responses: Thanks for the reviewer’s comment. We provided a summary of our results and added it to the first paragraph in the “Discussion and conclusions” section (Page 21, lines 15-18). 

-Is it possible to have an overview on each country data and how they correlate with CPS/IBSI and science achievements? Is there any outlier in the analysis?

Response: We agree with the reviewer that including country-specific results would provide more details to the readers; however, given that 52 countries were included in our study, presenting results for each country in this manuscript could be overwhelming to readers. 

Regarding outliers, we conducted outlier checks for each of the outcome variables by selecting 3 plausible values (PVs) out of the available 10 PVs and using histograms, z-scores, and Mahalanobis Distance as diagnostic tools. We did not include the outlier information in our manuscript because outliers did not affect our results. 

In summary, the histograms (please see attached in the word document labeled 'Response to Reviewers') indicated that the outcome variables exhibited approximately normal distributions. The z-scores and Mahalanobis Distance measures showed some potential outliers. We conducted analyses with and without outliers, and the conclusions remained consistent in both scenarios. So we decided to use the full sample size, which would be more representative. 

-It is sometimes hard to follow how raw data were converted into the data presented in the models. Mainly to the ones who do not know about statistics but have interest in this subject and in science education.

Response: We did not provide a detailed description of how raw data was converted, because data transformation was conducted by PISA researchers and the process has been documented in detail within the PISA reports. However, we have incorporated additional sentences in the Methods section (Page 10, lines 6-8, lines 11-14; Page 11, lines 10-11; Page 13, lines 8-10) and the Supporting Information file S1 (Page 2) to enhance clarity for readers. 

-Do the authors think that CPS could have impact in other disciplines?

Response: That is a great question. We have now added two sentences regarding the potential impacts of CPS in other disciplines (Page 24, lines 1-4). 

-It is not very clear why there are differences between valuing relationships and valuing teamwork to outcomes of the present investigation. They seem to correlate to each other.

Response: Thanks for the reviewer’s comment. We have now incorporated the correlation between valuing relationships and valuing teamwork into the manuscript. These two dimensions have a moderate correlation (r = 0.45), suggesting that they are different in terms of their conceptualization (Page 22, lines 8-12). While we did provide some initial explanations for this finding (Page 22, lines 11-21), we acknowledge that more in-depth research is needed to delve further into these relationships. 

Reviewer #2: 

The authors have done a fully revision of the manuscript. The authors have attended to the concerns highlighted by the reviewers. Although the authors have not provided the contents of the REU program, they addressed the Vygotsky's social constructivism theory.

The study was done with PISA 2015 data. Using the multilevel data the study aimed to investigate the relationship between collaboration dispositions and students’ collaborative problem-solving (CPS) competency. Also, CPS competency and inquiry-based science instruction were related to science achievement. Some novel data were highlighted by controlling some of the multilevel factors. For example, the study unveiled an important data that was masked by the HDI (countries indexes) effect, so that for some countries female students scores are higher than male students scores, showing that female students have more CPS competency than males.

Responses: Thank you for taking the time to review our manuscript and offering valuable feedback. The comments you provided before have been immensely helpful in improving our manuscript, and we appreciate your continued review of our work.

---

## [Decision Letter · Decision Letter 1]

27 Nov 2023

Investigating student collaborative problem-solving competency and science achievement with multilevel modeling: Findings from PISA 2015

PONE-D-23-22154R1

Dear Dr. Tang,

We’re pleased to inform you that your manuscript has been judged scientifically suitable for publication and will be formally accepted for publication once it meets all outstanding technical requirements.

Kind regards,

Sonia Vasconcelos, PhD

Academic Editor

PLOS ONE

Reviewers' comments:

Reviewer's Responses to Questions

**Comments to the Author**

1. If the authors have adequately addressed your comments raised in a previous round of review and you feel that this manuscript is now acceptable for publication, you may indicate that here to bypass the “Comments to the Author” section, enter your conflict of interest statement in the “Confidential to Editor” section, and submit your "Accept" recommendation.

Reviewer #1: All comments have been addressed

2. Is the manuscript technically sound, and do the data support the conclusions?

Reviewer #1: Yes

3. Has the statistical analysis been performed appropriately and rigorously? 

Reviewer #1: Yes

4. Have the authors made all data underlying the findings in their manuscript fully available?

Reviewer #1: Yes

5. Is the manuscript presented in an intelligible fashion and written in standard English?

Reviewer #1: Yes

6. Review Comments to the Author

Reviewer #1: The authors have implemented all suggestions raised by the reviewers. Regarding the data for each country (52 in total) the authors could not uncouple theses data at the present moment and the reasons for that were presented.

7. PLOS authors have the option to publish the peer review history of their article (what does this mean?). If published, this will include your full peer review and any attached files.

Reviewer #1: **Yes: **Debora Foguel

Reviewer #2: All comments have been addressed

 **********

2. Is the manuscript technically sound, and do the data support the conclusions?

Reviewer #2: Yes

 **********

3. Has the statistical analysis been performed appropriately and rigorously?

Reviewer #2: Yes

 **********

4. Have the authors made all data underlying the findings in their manuscript fully available?

Reviewer #2: Yes

 **********

5. Is the manuscript presented in an intelligible fashion and written in standard English?

Reviewer #2: Yes

 **********

6. Review Comments to the Author

Reviewer #2: As the raw data comes from PISA, the authors addressed only some statistical concerns highlighted by one of the reviewers. But they made the the text clearer.

 **********

7. PLOS authors have the option to publish the peer review history of their article (what does this mean?). If published, this will include your full peer review and any attached files.

**Do you want your identity to be public for this peer review?** For information about this choice, including consent withdrawal, please see our Privacy Policy.

Reviewer #2: **No**

---

## [Editor Report · Acceptance letter]

30 Nov 2023

PONE-D-23-22154R1 

Investigating student collaborative problem-solving competency and science achievement with multilevel modeling: Findings from PISA 2015 

Dear Dr. Tang:

I'm pleased to inform you that your manuscript has been deemed suitable for publication in PLOS ONE. Congratulations! Your manuscript is now with our production department. 

Kind regards, 

on behalf of

Dr. Sonia Vasconcelos 

Academic Editor

PLOS ONE